# Cilomilast Modulates Rhinovirus-Induced Airway Epithelial ICAM-1 Expression and IL-6, CXCL8 and CCL5 Production

**DOI:** 10.3390/ph17111554

**Published:** 2024-11-20

**Authors:** Jie Zhu, Michael R. Edwards, Simon D. Message, Luminita A. Stanciu, Sebastian L. Johnston, Peter K. Jeffery

**Affiliations:** 1Airway Disease, National Heart and Lung Institute, Imperial College London, Norfolk Place, London W2 1PG, UK or jiezhu57@gmail.com (J.Z.); michael.edwards@imperial.ac.uk (M.R.E.); s.johnston@imperial.ac.uk (S.L.J.); 2Thoracic Medicine, Gloucestershire Hospitals NHS Foundation Trust, Alexandra House, Sandford Road, Cheltenham GL53 7AN, UK; simon.message@glos.nhs.uk

**Keywords:** cilomilast, PDE4 inhibitors, rhinovirus, bronchial epithelial cells, ICAM-1, IL-6, CXCL8, CCL5

## Abstract

**Background:** Cilomilast, a phosphodiesterase-4 (PDE4) selective inhibitor, has anti-inflammatory effects in vitro and in vivo and reduces COPD exacerbations. We tested the hypothesis that cilomilast inhibits virus-induced airway epithelial intercellular adhesion molecule-1 (ICAM-1) expression and inflammatory cytokine/chemoattractants, IL-6, CXCL8, and CCL5 production in vitro. **Methods:** BEAS-2B bronchial epithelial cells were incubated with 0.5–2 MOI (multiplicity of infection–infectious units/cell) of rhinovirus 16 (RV16). Then, 0.1–10 μM cilomilast or 10 nM dexamethasone, as inhibition control, were added pre- or post-1 h RV16 infection. Supernatant and cells were sampled at 8, 24, 48, and 72 h after infection. Cell surface ICAM-1 expression was detected by immunogold labelling and visualised by high-resolution scanning electron microscopy (HR-SEM), while IL-6, CXCL8, and CCL5 protein release and mRNA expression were measured using an ELISA and RT-PCR. **Results:** Cilomilast significantly decreased RV16-induced ICAM-1 expression to approximately 45% (*p* < 0.01). CXCL8 protein/mRNA production was reduced by about 41% (*p* < 0.05), whereas IL-6 protein/mRNA production was increased to between 41–81% (*p* < 0.001). There was a trend to reduction by cilomilast of RV16-induced CCL5. **Conclusions:** Cilomilast has differential effects on RV16-induced ICAM-1 and interleukins, inhibiting virus-induced ICAM-1 expression and CXCL8 while increasing IL-6 production. These in vitro effects may help to explain the beneficial actions of this PDE4 inhibitor in vivo.

## 1. Introduction

Asthma and chronic obstructive pulmonary disease (COPD) are both inflammatory conditions of the airways that contribute significantly to global disease and financial burden [1,2]. Asthma and COPD exacerbations are a major cause of morbidity and mortality [3,4]. Their exacerbations are frequently linked to respiratory viral infections [5,6]. Human rhinovirus (RV) infections account for 60–65% of viral exacerbations in asthma [7,8] and COPD [9,10]. Direct examination of bronchial tissue by endobronchial biopsy has demonstrated distinct patterns of inflammatory cell recruitment in stable asthma and COPD, i.e., a predominance of CD4+ T cells and eosinophils in asthma and CD8+ T cells and macrophages in COPD [11]. In contrast, in both asthma and COPD, naturally occurring and RV-induced experimental exacerbations are associated with an increase in tissue neutrophils as well as eosinophils [12,13,14,15,16], also of interleukin-8 (IL-8,CXCL8) [12,15] and CC-type chemokine ligand 5 (CCL5) gene expression [14]. In exacerbations of mild COPD, there is a positive correlation between CCL5 and the extent of tissue eosinophilia [14]. Tissue recruitment and accumulation of specific subtypes of inflammatory cells is the result of selective, time-dependent up-regulation of cell surface adhesive molecules and cytokines/chemoattractants on both inflammatory and structural cells, including airway epithelium [17,18]. RV infection induces the expression of pro-inflammatory molecules in airway epithelium, including intercellular adhesion molecule-1 (ICAM-1) [19], interleukin-6 (IL-6), CXCL8 [20], and CCL5 [21]. Thus, modulation of RV-induced bronchial epithelial pro-inflammatory mediators and adhesion molecule production represents a potentially important therapeutic target and approach to the anti-inflammatory treatment of asthma, COPD, and possibly other virally induced lung inflammatory conditions. However, little is known about the capacity of potential new treatments to modulate such expression and production by bronchial epithelium.

The enzyme phosphodiesterase-4 (PDE4) normally reduces intracellular cyclic ade nosine-3′,5′-monophosphate (cAMP) by hydrolysis [22]. Cilomilast is one of several selective PDE4 inhibitors that increase intracellular cAMP with resultant suppression of airway inflammation [22]. It is already known that cilomilast is able to inhibit tumour necrosis factor-α (TNF-α) production by bronchial epithelial cells isolated from patients with COPD [23] and has the capacity to attenuate lipopolysaccharide (LPS)-induced pulmonary neutrophilia [24]. Several randomised, placebo-controlled trials have demonstrated that PDE4 inhibitors, such as cilomilast and roflumilast, improve lung function, decrease the risk of exacerbation [25,26], and even reduce the frequency of exacerbations requiring systemic corticosteroid therapy in patients with severe COPD [26]. A randomised placebo-controlled study of stable COPD demonstrated that cilomilast reduces the numbers of bronchial tissue CD8+ (T-cytotoxic) and CD68+ (macrophages) inflammatory cells, cells considered to be key players in the pathogenesis of COPD [27]. However, the underlying mechanisms of PDE4 inhibitors’ anti-pulmonary inflammatory effect are unclear. We wondered whether PDE4 inhibitors such as cilomilast affect cell surface adhesive molecule expression and cytokine/chemoattractant production, which are important to tissue recruitment of mononuclear and polymorphonuclear inflammatory cells in response to respiratory virus infection.

Thus, we hypothesised that cilomilast would inhibit RV-induced ICAM-1 expression and the production of IL-6, CXCL8, and CCL5 from the bronchial epithelium. To test this hypothesis, we designed experiments to evaluate the effects of cilomilast on a common cold virus RV16-induced adhesion molecule expression and cytokine/chemokine production on/from human bronchial epithelial cells in vitro. Such data may help to explain the reported anti-inflammatory effects of selective PDE4 inhibitors in asthma and COPD [25,26,27] and in response to respiratory viral lung infections.

## 2. Results

### 2.1. Effects of Cilomilast on RV16-Induced Surface ICAM-1 Expression

HR-SEM revealed that there were microvilli on the cell surface (Figure 1a) and showed constitutive ICAM-1 cell surface labelling (i.e., numbers of immuno-gold particles) on the non-infected BEAS-2B cells (Figure 1b). There were significantly more immuno-gold particles on the surface of BEAS-2B cells at 24 h after 50 µL (1 MOI) RV16 infection (Figure 1c) compared to the medium. However, exposure of epithelial cells to cilomilast attenuated RV16-induced ICAM-1 labelling (Figure 1d,e). As a negative control for the immunogold labelling procedure, RV16-infected BEAS-2B cells that had been labelled with an irrelevant antibody (MOPC21) had little or no ICAM-1 cell surface labelling (Figure 1f). ICAM-1 immuno-gold labelling on each cell was evenly distributed across the cell surface, both at baseline and after treatment.

The results of the quantification are presented as a frequency distribution histogram together with the median and range in Figure 2a–d. For non-infected and non-cilomilast treated BEAS-2B cells, approximately 80% of cells had less than 20 immuno-gold particles per 10 μm^2^ cell surface with a median (range) value = 7 (0–86) (Figure 2a). In contrast, RV16 infection significantly increased ICAM-1 gold particles, i.e., only 27% of infected BEAS-2B cells had less than 20 particles per 10 μm^2^ surface, and 53% of cells had up-regulated ICAM-1 expression with a median of 29 (0–240), (*p* < 0.0001 vs. medium only, Figure 2b). Moreover, 1 and 5 µM pre-cilomilast significantly inhibited the RV16-induced ICAM-1 expression, i.e., cells with less than 20 particles per 10 μm^2^ surface were increased to 45% with values as 18 (0–160) and 16 (0–158) (*p* < 0.01 vs. RV16 group, Figure 2c and 2d), respectively.

The data representing counts of 300 cells per group are summarised in Figure 3a,b and Table 1. Compared with the medium only, UV light inactivated and filtered RV16 did not induce ICAM-1 expression (Figure 3a). The non-virus containing 0.1% DMSO infection medium for dissolving the drug did not change and did not affect ICAM-1 baseline expression. RV16-infected BEAS-2B cells labelled with MOPC21 (the irrelevant antibody) had median values of 1.3, and no cell had more than 5 particles per 10 µm^2^ surface (Figure 3a). The non-specific labelling was deducted in the calculation procedure. RV16 infection alone significantly up-regulated cell surface ICAM-1 expression more than three-fold vs. non-RV16/UVRV16 groups (*p* < 0.0001 or <0.001, Figure 3a and Table 1). Both pre- and post-cilomilast treatment significantly suppressed RV16-induced ICAM-1 expression by 38–45%. A significant reduction at 1 μM cilomilast was detected in the pre- but not in the post-groups. Similar levels for both pre- and post-cilomilast could be seen with 5 and 10 μM cilomilast (*p* < 0.01 vs. RV16 only, Figure 3b and Table 1). Compared to RV16, ICAM-1 expression was up-regulated 3-fold higher than RV6 alone by a mix of TNF-α + IFN-γ, which acted as a positive control of up-regulation (*p* < 0.0001, Figure 3b and Table 1). As expected, Dex had a significantly bigger effect on inhibition of RV16-induced ICAM-1 expression to a 3-fold lower level than RV16 alone and, as such, acted as inhibition control (*p* < 0.0001 vs. RV16 group, Figure 3b and Table 1). In addition, cilomilast significantly reduced the baseline level of ICAM-1 expression (*p* < 0.05 vs. non-RV16 group, Table 1).

The coefficient of variation (% CV) for five repeat counts of the same cell surface area was 1.6%.

### 2.2. Time and Dose Course of RV16 Infection on IL-6, CXCL8, and CCL5 Protein Production: Relation to Effect of Cilomilast Treatment

Compared to the medium, UV-inactivated RV16, filtered RV16, and the non-virus 0.1% DMSO infection medium did not significantly induce IL-6, CXCL8, or CCL5 protein release (Figure 4). After exposure to RV16, the concentrations of IL-6, CXCL8, and CCL5 (*p* < 0.05) protein in BEAS-2B cell supernatant were elevated at 8–72 h in a time-dependent manner (*p* < 0.001 for IL-6 and CXCR8; *p* < 0.01 for CCL5, Figure 4).

The 25–100 µL (0.5–2 MOI) RV16 infection significantly increased IL-6 protein production to similar levels (*p* < 0.001, Figure 5a). Surprisingly, compared with RV16 alone, 1–10 µM cilomilast significantly augmented RV16-induced IL-6 protein production from 24 h to 72 h in both pre-treated (Figure 5b) and post-treated groups (*p* < 0.01). The effect increased with time in a cilomilast dose-dependent manner (Figure 5a,b). Compared with RV16 infection alone, IL-6 protein production was increased by cilomilast 2–3 folds. In contrast, 10 nM Dex significantly decreased RV16-induced IL-6 protein production at pre-treated 24 h and 48 h time points (Figure 5b), also at the post-treated 48 h time-point (*p* < 0.01 vs.RV16 alone). Moreover, compared to medium alone, 1–10 µM cilomilast significantly increased baseline IL-6 protein by 2 folds at 8 h or beyond (*p* < 0.01 vs. medium, Figure 5c).

Adding 25–100 µL (0.5–2 MOI) RV16 significantly induced CXCL8 protein production in both the virus dose- and infection time-dependent manner (*p* < 0.001, Figure 6a,b). In contrast to IL-6, a significant reductive effect of cilomilast was detected at 25, 50, and 75 µL but not at 100 µL RV16 groups (*p* < 0.05, Figure 6a). Compared with RV16 alone, RV16-induced CXCL8 production was significantly down-regulated by 25–36% after pre-1–10 µM cilomilast (Figure 6a,b) and post-5–10 µM cilomilast treatment but only at the 24 h time-point (*p* < 0.01). Both pre- (Figure 6b) and post-10 nM Dex significantly reduced RV16-induced CXCL8 protein production by 72–78% from 8 to 24 h (*p* < 0.01) and by 31% at 48 h (*p* < 0.05), but not at 72 h time point. Compared to medium alone, cilomilast at 5 µM and 10 µM significantly reduced baseline CXCL8 production by 19–25% at 8 h (*p* < 0.05, Figure 6c), and there was a trend toward a decrease in the 10 µM cilomilast treatment at 24 h (*p* = 0.06).

Adding 25–100 µL (0.5–2 MOI) RV16 significantly induced CCL5 protein production in a virus dose- and infection time-dependent manner (*p* < 0.01, Figure 7a,b). However, both pre- and post- cilomilast had only a small, non-significant reductive effect on RV16-induced CCL5 protein production at 48 h RV16 infection (Figure 7b). In contrast, both pre- (Figure 7b) and post-10 nM Dex significantly inhibited CCL5 production at the 48 and 72 h time points (*p* < 0.05). Moreover, 0.1–10 µM cilomilast did not significantly inhibit baseline CCL5 protein release.

### 2.3. Effects of Cilomilast on RV16-Induced IL-6, CXCL8, and CCL5 mRNA Synthesis

In comparison to the non-infected control, RV16 significantly up-regulated IL-6 (*p* < 0.01, Figure 8a) and CXCL8 mRNA synthesis in a time-dependent manner (*p* < 0.001, Figure 8b). CCL5 mRNA was increased at 24 h and peaked at 48 h after RV16 infection (*p* < 0.001, Figure 8c). In agreement with the results for protein, 1–10 µM cilomilast treatment significantly increased IL-6 mRNA copy numbers at 24, 48, and 72 h in a drug dose-dependent manner. In contrast, 10 nM Dex significantly reduced IL-6 mRNA production at 24–72 h (*p* < 0.05 or *p* < 0.01, Figure 8a). Pre-1–10µM cilomilast significantly decreased RV16-induced CXCL8 mRNA at both 8 and 24 h (*p* < 0.05, Figure 8b) but not at 48 and 72 h. There was a non-significant trend toward a reduction in RV16-induced CCL5 mRNA synthesis with pre-1–10µM cilomilast at 24–72 h (Figure 8c). Compared with RV16 only, 10 nM Dex significantly inhibited CXCL8 at all time points and CCL5 mRNA at 24–72 h (*p* < 0.01) with a greater degree of reduction than all dosages of cilomilast (Figure 8a–c). Compared to the non-virus controls, UV light-inactivated RV16 and filtered RV16 did not induce IL-6, CXCL8, and CCL5 mRNA synthesis. Non-virus medium with the included 0.1% DMSO used for dissolving the drug also did not affect IL-6, CXCL8, or CCL5 mRNA production.

There was little or no difference between pre-treatment and post-treatment. However, as we described in the above result Section 2.1, ‘A significant reduction at 1 μM cilomilast was detected in the pre, but not in the post groups’, and in 2.2 as ‘RV16 induced CXCL8 production was significantly down-regulated by 25–36% after pre-1–10 µM and post-5–10 µM cilomilast treatment but only at the 24 h time-point’. These results demonstrate that pre-treatment is more effective at the lower dosage of cilomilast than post-treatment.

## 3. Discussion

The present study of a much-used bronchial epithelial cell line demonstrates for the first time that cilomilast, a selective second-generation PDE4 inhibitor, decreases RV16-induced ICAM-1 cell surface expression and CXCL8 release while increasing IL-6 production. There was only a small, non-significant reductive trend on RV16-induced CCL5. ICAM-1 or CD54 is a 90-KD, inducible surface glycoprotein and one of the major ligands for lymphocyte function-associated antigen-1 (LFA-1) and macrophage 1 antigen (Mac-1; CD11b/CD18, αMβ2) integrin family, expressed on the surface of leukocytes [28,29]. Epithelial ICAM-1 has a dual role as the major receptor for RV16 and as a cell–cell adhesion molecule for inflammatory cells [19]. Up-regulation of ICAM-1 is, thus, important in common cold virus infection, also in margination, migration, and retention of circulating neutrophils, monocytes, lymphocytes, and eosinophils in airway and lung tissues [17,18]. Several studies have shown that airway inflammation is characterised by an up-regulation of ICAM-1 on airway epithelium, which is of particular importance to the induction of antigen or viral-induced airway inflammation [30,31]. Previously, the PDE4 inhibitor roflumilast was shown to inhibit LPS-induced endothelial P- and E-selectin expression and suppress neutrophil adhesion to TNF-α-activated human umbilical vein endothelial cells in vitro [32]. Moreover, pharmacological agents such as forskolin and a β2 agonist, procaterol, increased the intracellular levels of cAMP, which inhibited TNF-α, interleukin-1 beta (IL-1β), and RV14 infection-induced ICAM-1 expression in cultured human smooth muscle cells [33] and bronchial epithelial cells [34], respectively. As histo/cyto pathologists, we tested the hypothesis and questions therein using our known expertise and experience with immuno-gold labelling and scanning electron microscopy [35]. Therefore, we could uniquely visualise and quantify cell surface ICAM-1 expression on individual bronchial epithelial cells. We considered it a more sensitive technique to detect cell surface ICAM-1 expression rather than Western blot, PCR, and ELISA, as these measure intracellular and soluble levels. We have demonstrated here, for the first time, that cilomilast significantly down-regulates both the baseline levels and RV16-induced surface ICAM-1 expression on bronchial epithelial cells. The in vitro findings suggest that PDE4 inhibitor reduction of surface ICAM-1 expression may explain the suppression of viral-associated retention and accumulation of inflammatory cells in vivo.

IL-6 was originally identified as a T cell-derived lymphokine that induces the final maturation step of B cells into antibody-producing cells. IL-6 and its receptor are expressed throughout the body in a wide variety of cell types. Historically, they are best known for their involvement in immune-mediated responses to infection, trauma, or injury [36,37]. Subsequent and recent studies reveal that IL-6 possesses pleiotropic activities that play a central role in regulating a plethora of cellular processes in the body, such as proliferation, differentiation, and functional maturation [37]. IL-6’s wide variety of activity in a range of autoimmune, inflammatory, and infectious diseases has led to various therapeutic interventions being successfully used in the clinic to treat conditions such as rheumatoid arthritis [37].

Although the role of IL-6 in inflammatory lung conditions, such as COPD, has not yet been clearly defined, its concentration in sputa [38,39] and serum [40,41] of patients with COPD is significantly elevated and associated with disease severity [38,39]. In COPD, where there are frequent (virally associated) exacerbations, sputum IL-6 concentration is related to higher numbers of sputum total cells, eosinophils, and lymphocytes [38].

In contrast to CXCL8, we demonstrate herein that cilomilast, in the 1–5 µM therapeutic range, significantly elevates both constitutive and RV16-induced IL-6 production from bronchial epithelial cells in a dose-dependent manner. The finding contrasts with our initial assumption but is in accord with previous in vitro demonstrations that other cAMP-elevating agents (such as salmeterol, salbutamol, forskolin, and rolipram) induce increased IL-6 production by bronchial epithelial cells [42].

Why did cilomilast decrease levels of ICAM-1 and CXCL-8 but increase levels of IL-6? The IL-6 gene contains several elements for transcription factor binding, including cAMP-responsive elements binding protein (CREB), nuclear factor-κB (NF-κB), activator protein 1 (AP-1), and others [43]. There is increasing literature reporting that intracellular cAMP is increased by PDE4 inhibitor-regulated genes/proteins of pre-inflammatory mediates, including IL-6 [44]. The IL-6 response to viral infection appears to be mediated via NF-kB [42], whereas its enhancement by cAMP-elevating agents is via a mechanism dependent upon cAMP response element cis-acting sites [42]. In contrast, activation of ICAM-1 and CXCL8 gene/protein is mediated via an NF-κB and AP-1 signalling pathway [45,46]. PDE4 inhibitors increase intracellular cAMP levels, which activates protein kinase A (PKA) and activates PKA phosphorylates CREB, promoting and augmenting rhinovirus induction of IL-6 production [37,43,44]. In contrast, activated PKA suppresses the NF-κB pathway to inhibit the production of pro-inflammatory mediators such as ICAM-1 [45] and CXCL8 [46].

In addition, the suppressive effect of corticosteroids is mediated by distinct glucocorticoid response elements within the IL-6 promoter [42]. While the functional significance and consequences of such findings are unknown, there is evidence to suggest that increased concentrations of IL-6 may not necessarily be harmful in COPD. For example, after ischemia and reperfusion of the lungs, administration of intratracheal recombinant IL-6 reduces endothelial disruption and neutrophil accumulation [47]. Tomura and colleagues have shown that IL-6 dose-dependently suppresses IFN-γ production in mouse allogeneic spleen cell culture models, indicating that IL-6 may down-regulate cytotoxic T-lymphocyte responses via inhibition of Th1 cytokines [48]. This last-mentioned action may be of relevance to COPD, an inflammatory condition [1] in which CD8+ T-lymphocytes and macrophages predominate in the airway and alveolar tissues of smokers [11].

CXCL8 is a chemoattractant and activating cytokine produced by bronchial epithelial cells, endothelial cells, fibroblasts, alveolar macrophages, and neutrophils [20,46]. CXCL8 is released only under inflammatory conditions to recruit and activate neutrophils to release granule contents, causing tissue damage [46]. Neutrophils are able to release CXCL8 in response to TNF, IL-1, endotoxin, and virus [20,46]. Analyses of induced sputum from COPD subjects during their stable phase show that raised CXCL8 is associated with greater exacerbation frequency [38]. Moreover, in severe exacerbations of COPD [38] or asthma [49], there are significantly raised sputum CXCL8 concentrations and increased tissue recruitment of neutrophils [12,15]. These studies have revealed the pivotal roles of CXCL8 in recruiting neutrophils into inflammatory sites of lung tissue and suggested that CXCL8 could be a novel target to protect against neutrophil-associated lung damage. In vitro, Profita and colleagues have demonstrated that cilomilast significantly decreases neutrophil chemotaxis [23]. Moreover, in vivo, Spond and colleagues have shown that rats pre-treated with cilomilast display dose-dependent inhibition of LPS-induced neutrophilia, but the inhibitory activity is not associated with TNF-α or IL-1 inhibition [24].

In the present study, we have demonstrated for the first time that cilomilast significantly reduces CXCL8 mRNA and protein production by 25–75 µL (0.5–1.5 MOI) RV16-infected bronchial epithelial cells at 8 and 24 h, but not at 48 and 72 h. The inhibitory effects of cilomilast on 100 µL (2 MOI) RV16-mediated CXCL8 production were overcome by a relatively high dose RV16 infection. Moreover, cilomilast significantly reduced baseline CXCL8 mRNA synthesis and protein release at 8 and 24 h drug treatment, respectively, but not at 48 and 72 h. Our findings indicate that cilomilast may have the capacity to attenuate neutrophil accumulation at bronchial sites of acute virus infection, at least partially, via inhibition of CXCL8 production.

Finally, CCL5/RANTES (regulated on activation normally T cell expressed and secreted) is considered to be released from a variety of cells, such as bronchial epithelial cells [14,21] and activated inflammatory cells [50]. CCL5 plays diverse roles in the pathogenesis of many inflammatory conditions [50,51]. Its chemotaxis activity recruits immune cells, including monocytes, T cells, NK cells, eosinophils, mast cells, basophils, and dendritic cells, to sites of inflammation and infecting pathogens [51,52,53,54]. Therefore, CCL5 is associated with a wide range of immune-mediated diseases, including asthma, airway inflammatory disorders, pathogen infection, rheumatoid arthritis, and atopic dermatitis [51]. A recent publication indicates that CCL5 is a potential bridge between type 1 and type 2 inflammation in asthma [55]. PDE4 inhibitors have been shown to inhibit eosinophil PDE4 activity in vitro and eosinophil recruitment in the lungs and skin in response to antigen challenge and a range of stimuli [56]. However, we showed that cilomilast has only a small, non-significant reductive effect on RV16-induced bronchial epithelial CCL5 protein and mRNA production.

We acknowledge that a weakness of this study is that we did not measure virus load in the cilomilast plus rhinovirus experiments. It is possible that cilomilast may have reduced ICAM-1 and CXCL8 induction by rhinovirus by suppressing rhinovirus replication per se. This seems unlikely, as we are not aware of data suggesting that PDE4 inhibitors have antiviral activity. There is one report demonstrating that the PDE4 inhibitor piclamilast had no effect on rhinovirus replication in either primary bronchial epithelial cells or airway smooth muscle cells [57]. Moreover, IL-6 induction by rhinovirus was further augmented by cilomilast, not reduced. The possibility that cilomilast may somehow suppress rhinovirus replication should be the subject of future studies. We will measure the viral load in the future when viral infection studies are performed.

In conclusion, the present in vitro data are compatible with the hypothesis that cilomilast decreases RV16-induced expression of ICAM-1 and CXCL-8 while increasing levels of RV16-induced IL-6 on or from bronchial epithelial cells. The clinical significance of PDE4 inhibitor-induced IL-6 requires further investigation.

Significantly, the present data highlight cellular mechanisms likely to be involved in the clinical effects of PDE4 inhibitors on the treatment of asthma and COPD [58] and help to explain their role in the prevention of virally trigged exacerbations of asthma and COPD and other viral infection-induced lung conditions.

## 4. Materials and Methods

### 4.1. BEAS-2B Cells and Virus Culture

We used BEAS-2 B cells, a bronchial epithelial cell line derived from normal human bronchial epithelial cells transfected with an adenovirus (Ad) 12 and SV40 hybrid virus. These have unlimited proliferative potential, but no tumours form [59]. Our BEAS-2B cells were obtained from the European Collection of Cell Cultures (ECACC 95102433). Cells were grown in RPMI 1640 media supplemented with Glutamax (Invitrogen, Waltham, MA, USA) with 10% fetal calf serum (FCS, Invitrogen) buffered with 1% sodium bicarbonate (Invitrogen) and 0.075% HEPES (Invitrogen). Cells were grown at 37 °C in a humidified incubator using 175-cm^2^ flasks and split when confluent. BEAS-2B cells were used at passages 4–10 in the experiments.

The identity of RV serotype 16 (RV16), a major RV subtype, was confirmed by neutralisation using serotype-specific antibodies. RV16 was grown in Ohio HeLa cells and titrated in confluent HeLa cells to ascertain a 50% tissue culture infectious dose (TCID_50_)/mL, as described [59]. The stock of RV16 was 2 × 107 TCID_50_/mL. In order to assess the specificity of RV-mediated responses, RV16 stock was exposed to either UV light as previously described [60] or filtered through a 30-kDa membrane (Millipore, Stonehouse Gloustershire, UK) at 10,000× *g* in a microcentrifuge to exclude virus.

### 4.2. Reagents

Cilomilast (SB-207499, GlaxoSmithKline, Philadelphia, PA, USA) was dissolved in 1% dimethyl sulfoxide (DMSO) infection medium (RPMI 1640 medium + 2% FCS), filtered through 0.2 µm filter, and diluted in 0.1% DMSO infection medium before use. Dexamethasone (Dex) sodium phosphate injection 8 mg/2 mL (2101B, David Bull Laboratories, Warwick, UK) was diluted in the infection medium. Human recombinant (hr) tumour necrosis factor-α (TNF-α) and hr interferon-γ (IFN-γ) (R&D systems, Abingdon, UK) were diluted in the infection medium.

### 4.3. Infection and Treatment

BEAS-2B cells were seeded onto coverslips in 12-well plates at 1.2 × 105 cells/well for detecting cell surface ICAM-1 expression and were also seeded onto 12-well plates for detecting IL-6, CXCL8, and CCL5 protein and mRNA.

At 90% confluence, the following experiments were carried out:(1)For pre-cilomilast treatment groups, BEAS-2B cells were incubated with a 0.1–10 µM dose of cilomilast 1 h prior to various doses of 25–100 µL (0.5–2 MOI, multiplicity of infection–infectious units/cell) of RV16.(2)For post-cilomilast treatment groups, BEAS-2B cells were infected with 25–100 µL of RV16 for 1 h at 33 °C with gentle shaking, after which the virus-containing medium was removed, and the cells were washed. Virus-free media containing 0.1–10 µM cilomilast were added.

For subsequent experiments designed to investigate the time course of IL-6 and chemokines production, the addition of 50 µL (1 MOI) RV16 stock/well was used as the standard dose throughout to ensure that the effects of cilomilast on RV16-induced IL-6, CXCL8, and CCL5 expression could be reliably detected.

For the detection of ICAM-1, the cells were infected only with 50 µL (1 MOI) of RV16.

As inhibition controls for ICAM-1, IL-6, CXCL8, and CCL5, pre/post-10 nM Dex was added instead of cilomilast.

Moreover, as a positive control for induction of ICAM-1, the cells were treated with combined pre/post-TNF-α (10 ngmL−1) and IFN-γ (40 ngmL−1) without RV16.

In addition, to investigate the effect of cilomilast on baseline ICAM-1, IL-6, CXCL8, and CCL5, 1–10 µM cilomilast was added in the absence of RV16 infection.

For non-RV16 infection controls, 0.1% DMSO infection media, filtered and UV light inactivated 25–100 µL of RV16, diluted in 0.1% DMSO infection media, was added to replace RV16.

### 4.4. Immunogold Labelling ICAM-1

In the present study, we applied our previously validated high-resolution scanning electron microscopy (HR-SEM) fitted with a backscatter detector to quantify immunogold labelling ICAM-1 expression on the surface of the bronchial epithelium [35]. Briefly, the coverslips on which the cells were grown were collected only at 24 h after 1 MOI RV16 and cilomilast treatment. The BEAS-2B cells were incubated with RR1/1, a mouse anti-human ICAM-1 (CD54) monoclonal antibody (mAb) (a gift from R. Rothlein. Boehringer Ingelheim, Ridgefield, CT, USA). Mouse myeloma IgG_1−k_ (MOPC21) (Sigma, Poole, Dorset, UK) was used as the negative control Ab instead of ICAM-1 mAb. Goat anti-mouse 30 nm colloidal gold-conjugated Ab (Amersham Life Science, Amersham, UK) was used to label anti-ICAM-1 mAb. An HR-SEM (S-4000; Hitachi Scientific Instruments Nissei Sangyo Co., Ltd., Tokyo, Japan) fitted with a backscatter detector (K.E. Developments Ltd., Cambridge, UK) enabled us to observe and quantify immuno-gold particle distribution on the surface of the cells.

### 4.5. ELISA for IL-6, CXCL8 and CCL5 Proteins

The supernatants of the BEAS-2B cells were collected at 8, 24, 48, and 72 h time points after 1–2 MOI RV16 and drug treatment. IL-6, CXCL8, and CCL5 proteins in the supernatant were measured using a quantitative sandwich enzyme-linked immunosorbent assay (ELISA) (R&D Systems Europe, Abingdon, UK), according to the manufacturer’s recommendations. Briefly, monoclonal capture antibodies specific to IL-6, CXCL8, and CCL5 (R&D Systems, MAB 206, MAB 208, and MAB 678), respectively, were precoated on microtiter plates and 100 µL of supernatant was added in duplicate. After incubation with biotinylated detection antibodies specific to IL-6, CXCL8, and CCL5 (R&D Systems, BAF 206, BAF 208, and BAF 278), respectively, streptavidin HRP and a substrate solution were added for colour development, and the optical density of each well was read using a microplate reader at 450 nm and 540 nm wavelength. Compared to a standard curve, quantification of each sample was obtained. The sensitivity of the assay was 7 pg/mL.

### 4.6. RT-PCR for IL-6, CXCL8 and CCL5 mRNA

A total of 0.5 mL of TrizolTM was added directly onto BEAS-2B cell monolayers after removal of the supernatant and stored at −80 °C in the freezer for assay of mediator mRNA expression. Total RNA was extracted using a, Hilden, Germany). A total of 2 µg of RNA was used for cDNA synthesis (Omniscript RT kit, Qiagen, Hilden, Germany). Specific primers and probes of IL-6, CXCL8, CCL5, and 18S rRNA used for quantitative reverse transcription real-time-polymerase chain reaction (RT-PCR) are summarised in Table 2

Each reaction consisted of 12.5 μL 2× QuantiTect Probe PCR Master Mix (Qiagen), 900 nM sense and 900 nM antisense primers and 175 nM probe (CXCL8) or 300 nM sense and antisense and 175 nM probe (18S rRNA), and 2 μL cDNA (18S rRNA reactions used 2 μL of a 1/100 dilution of the cDNA) made up to 25 μL with nuclease-free water (Promega, Madison, WI, USA). The reactions were analysed using an ABI7500 Automated TaqMan (ABI, Foster City, CA, USA). The amplification cycle consisted of 50 °C for 2 min, 94 °C for 10 min, and 45 cycles of 94 °C for 15 s, 60 °C for 1 min. Data were analysed using ABI 7500 SDS V2.3 Software (ABI). IL-6, CXCL8, and CCL5 quantitative PCR data were normalised to 18S rRNA and presented as copies of IL-6, CXCL8, and CCL5 mRNA/µg of total RNA using a standard curve based on amplification with plasmid DNA.

### 4.7. Statistical Analyses

For ICAM-1 expression, the counting data of 300 cells per group from three coverslips of three experiments were analysed. The data for cell surface ICAM-1 expression were expressed as median (M) and range (R) gold particle number per 10 μm^2^ cell surface. As the data were non-normally distributed, the Kruskal–Wallis test was used to compare differences among multiple groups, and then the Mann–Whitney U-test was applied for differences between the two groups. For IL-6, CXCL8, and CCL5 protein and mRNA measurements of ELISA and RT-PCR, four experiments were carried out in duplicate. Therefore, the data of 8 samples per group were analysed. The mean and standard deviation (SD) were calculated in each case. The ELISA and RT-PCR data were expressed as mean ± SD. Comparisons of means were made using one-way ANOVA for differences between multiple groups, followed by the paired Student’s *t*-test for differences between the two groups. A *p*-value < 0.05 was accepted as indicating a statistically significant difference. GraphPad Prism 4.00 software was used.

## Figures and Tables

**Figure 1 pharmaceuticals-17-01554-f001:**
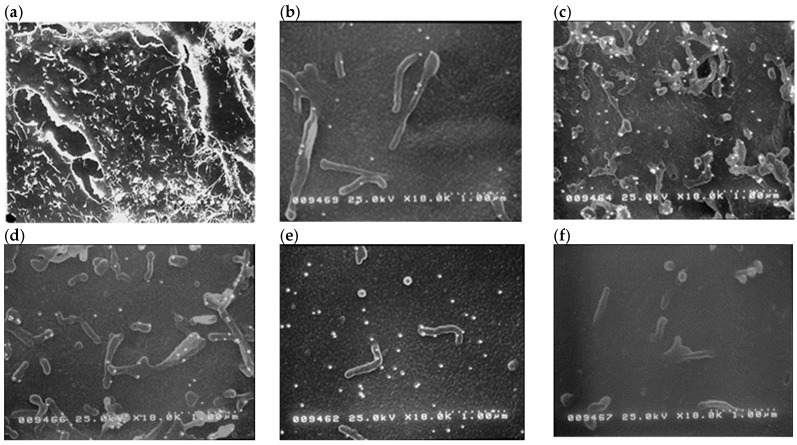
High-resolution scanning electron microscopy mixed backscatter and secondary electron images of immunogold-labelled adhesion molecule-1 (ICAM-1) on the surface of BEAS-2B cells: (**a**) Lower magnification showing an entire single cell with microvilli on its surface (Internal scale bars = 20 µm). (**b**–**f**) One counting field of higher magnification (Internal scale bars = 1 µm) showing (**b**) fewer gold particles on the surface of non-infected cells, (**c**) more gold particles on 50 µL (1 MOI) RV16-infected cells, (**d**) 1 µM and (**e**) 5 µM cilomilast pre-treatment reduced the numbers of gold particles on RV16-infected cells, (**f**) no particles on cells labelled with irrelevant mouse myeloma IgG_1−k_ (MOPC21) instead of ICAM-1 mAb.

**Figure 2 pharmaceuticals-17-01554-f002:**
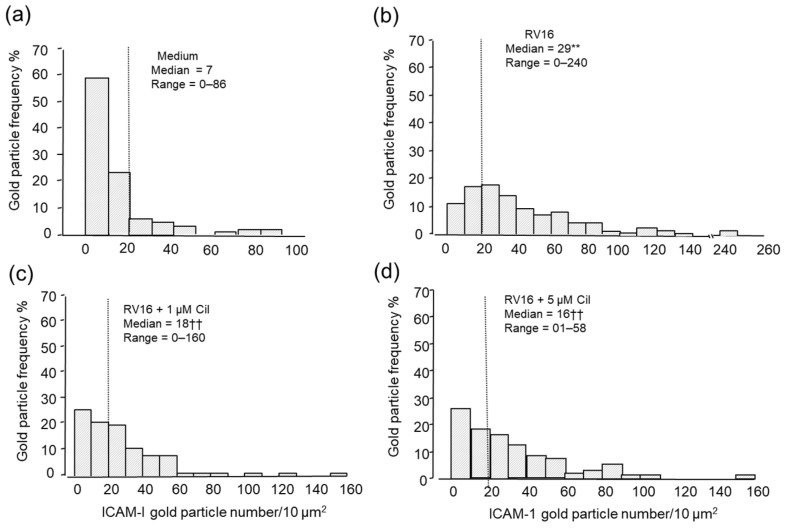
ICAM-1 gold particle frequency distribution of BEAS-2B cells assessed by high-resolution scanning electron microscopy. The results are expressed as the number of gold particles per 10 μm^2^ area surface against the relative frequency with medians and ranges as shown for each group. (**a**) Non-infected and (**b**) a significant increase in ICAM-1 expression after RV16 infection for 24 h, ** *p* < 0.01 versus (vs.) medium only, (**c**) 1 µM and (**d**) 5 µM cilomilast pre-treatment significantly reduced ICAM-1 expression on the surface of RV16-infected BEAS-2B cells, ^††^ *p* < 0.01 vs. RV16-infected cells (*n* = 300, Mann–Whitney U-test).

**Figure 3 pharmaceuticals-17-01554-f003:**
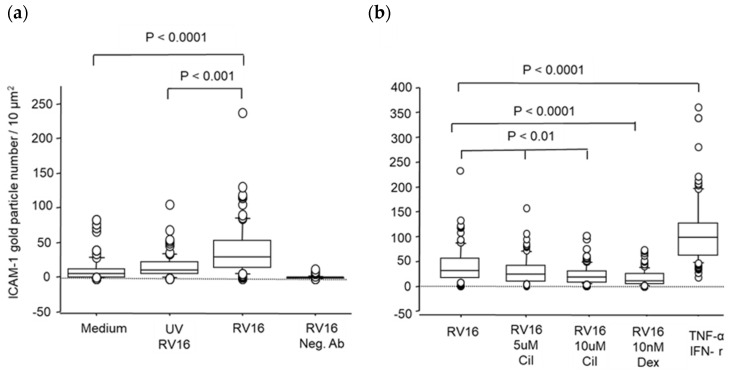
Box plot of comparisons of ICAM-1 expression on pre-treated BEAS-2B cells. Counts of ICAM-1 gold particles on BEAS-2B cells assessed by high-resolution scanning electron microscopy. The results are expressed as the median, interquartile range, and full range of gold particle numbers per 10 μm^2^ area of cell surface for each group. (**a**) Compared to medium only and UVRV16, RV16 infection significantly increased ICAM-1 gold particles at 24 h. (**b**) Compared to RV16, both 5 μM and 10 μM cilomilast significantly reduced gold particle number, and dexamethasone (Dex) had the biggest inhibition of RV16-induced ICAM-1 expression. As the positive control, ICAM-1 was up-regulated three-fold higher by a mix of TNF-α + IFN-γ. Results are median (range) (*n* = 300 cells, Mann–Whitney U-test).

**Figure 4 pharmaceuticals-17-01554-f004:**
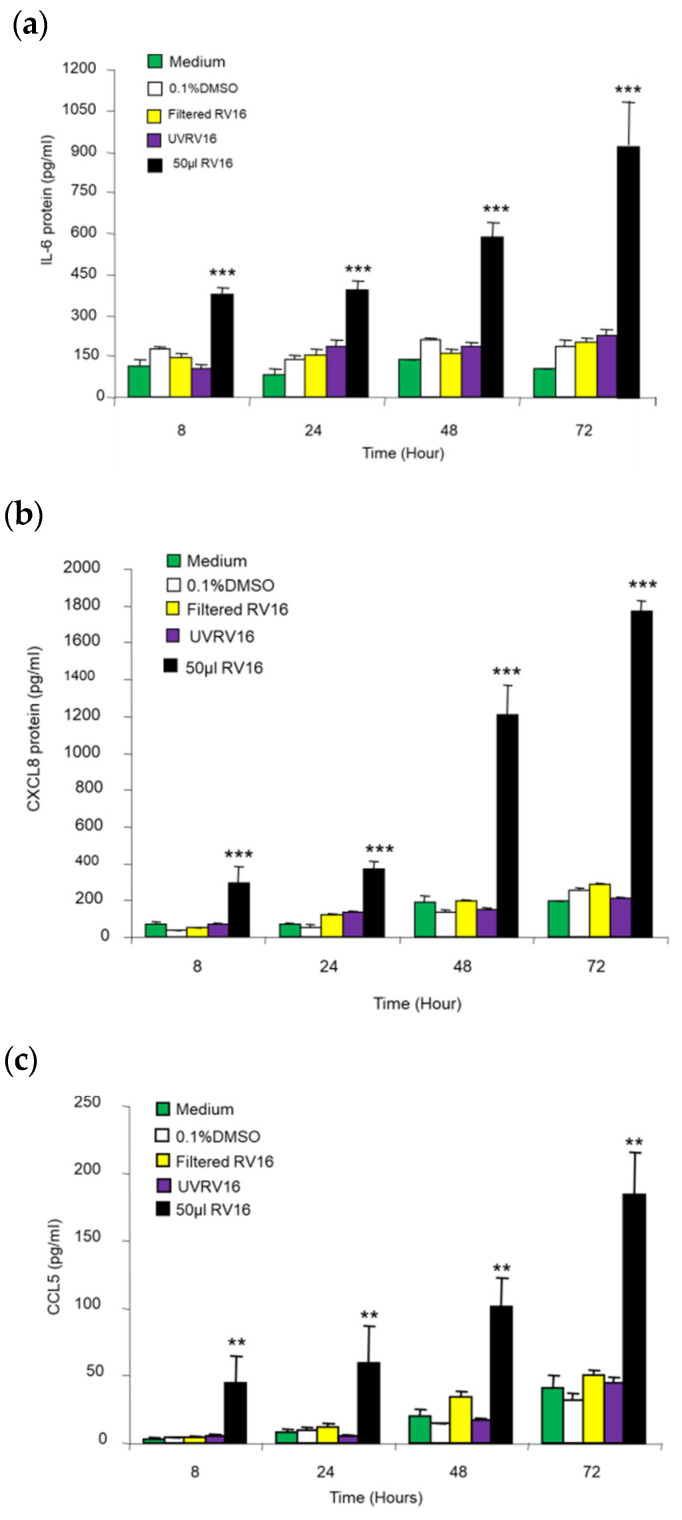
Time course of 50 μL (1 MOI) RV16 infection for induction of (**a**) IL-6, (**b**) CXCL8, and (**c**) CCL5 protein production from BEAS-2B cells. Exposure of RV16 medium to UV light (UV) or filtration through a 30-kDa filter almost completely abrogated RV16-mediated IL-6, CXCL8, and CCL5 up-regulation. ** *p* < 0.01 and *** *p* < 0.001 vs. medium, DMSO, filtered RV16, and UVRV16 groups, respectively. Results are mean ± SD (*n* = 8, paired Student’s *t*-test).

**Figure 5 pharmaceuticals-17-01554-f005:**
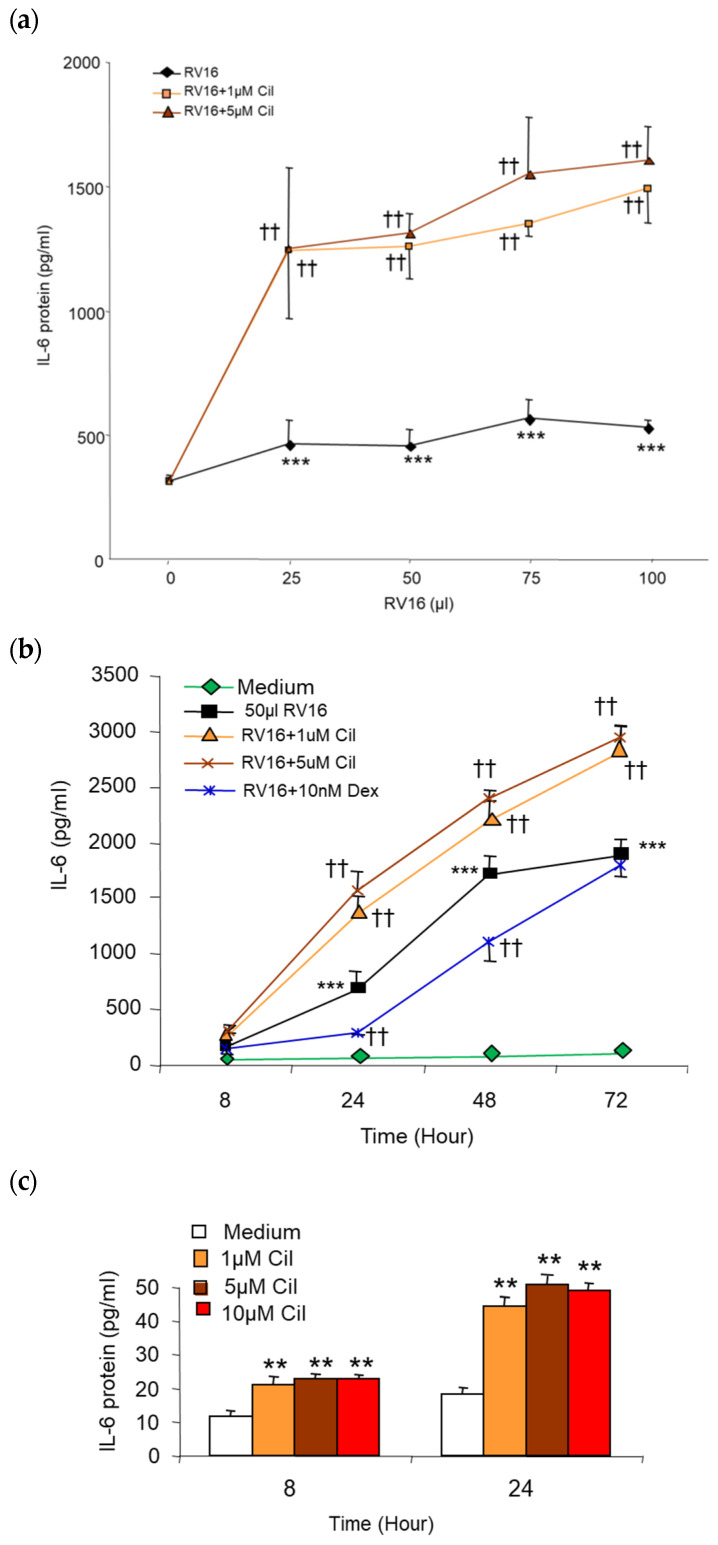
Effect of pre-cilomilast on IL-6 protein release from BEAS-2B cells (**a**) 25–100 µL (0.5–2 MOI) of RV16 infection at 24 h significantly induced IL-6 production to similar levels; also 1 and 5 µM cilomilast significantly enhanced the 25–100 µL RV16-induced IL-6 production in a drug dose-dependent manner. (**b**) The 1 and 5 µM cilomilast significantly increased 50 µL (1 MOI) RV16-induced IL-6 production in a time-dependent manner, whereas 10 nM Dex significantly decreased 50 µL RV16-induced IL-6 at 24 and 48 h. (**c**) The 1–10 µM cilomilast significantly increased baseline IL-6 production. ** *p* < 0.01 and *** *p* < 0.001 vs. medium group; ^††^ *p* < 0.01 vs. RV16 groups. Results are mean ± SD (*n* = 8, paired Student’s *t*-test).

**Figure 6 pharmaceuticals-17-01554-f006:**
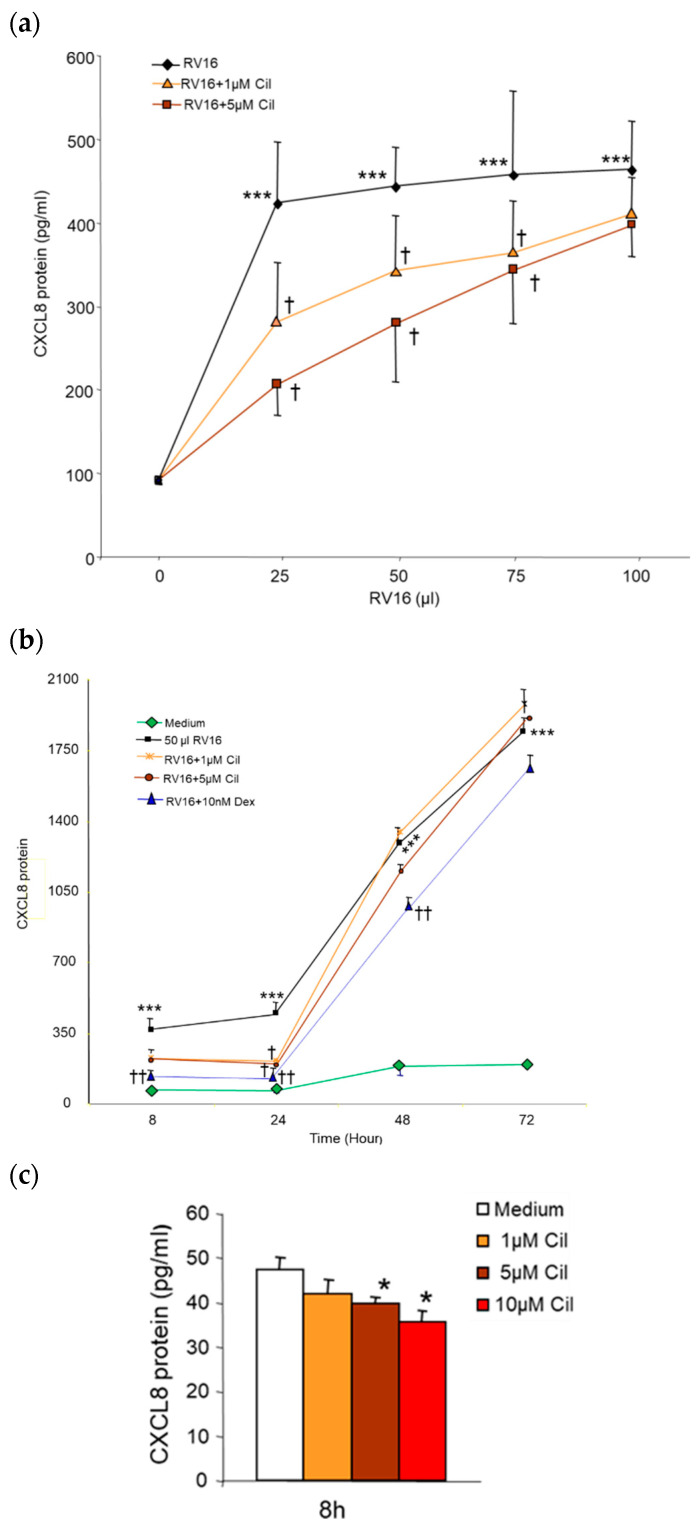
Effect of pre-cilomilast on CXCL8 protein production from BEAS-2B cells. (**a**) The 25–100 µL (0.5–2 MOI) of RV16 infection at 24 h significantly induced CXCL8 production to similar levels, and 1 and 5 µM cilomilast significantly reduced the 25–75 µL (0.5–1.5 MOI) RV16-induced CXCL8 production in a drug dose-dependent manner. (**b**) Time course of cilomilast on CXCL8 release from 50 µL (1 MOI) RV16-infected BEAS-2B cells. A significant reductive effect of 1 and 5 µM cilomilast was detected only at 24 h, and 10 nM Dex significantly decreased 50 µL RV16-induced CXCL8 from 8 to 48 h. (**c**) The 5 and 10 µM cilomilast (Cil) significantly inhibited constitutive CXCL8 production at 8 h. * *p* < 0.05 and *** *p* < 0.001 vs. medium; ^†^ *p* < 0.05 and ^††^ *p* < 0.01 vs. RV16 groups. Results are mean ± SD (*n* = 8, paired Student’s *t*-test).

**Figure 7 pharmaceuticals-17-01554-f007:**
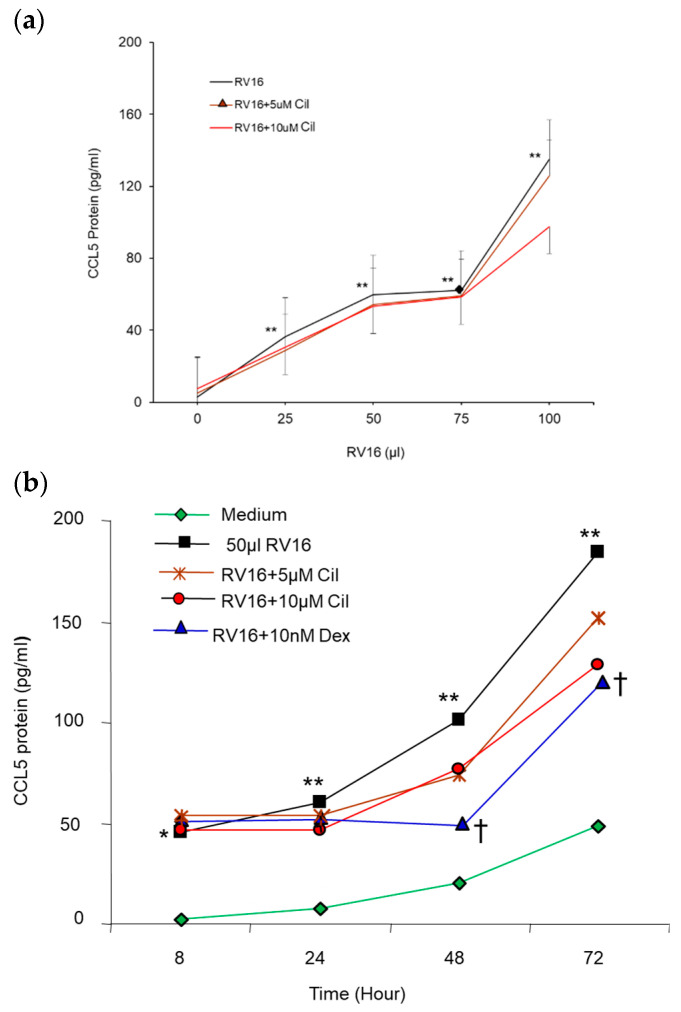
Effect of pre-cilomilast on CCL5 protein production from BEAS-2B cells. (**a**) The 25–100 µL (0.5–2 MOI) of RV16 infection at 24 h significantly increased CCL5 release in an RV16 dose-dependent manner. The 5 and 10 µM cilomilast doses did not inhibit RV16-induced CCL5 level. (**b**) Time course of CCL5 production from 50 µL (1 MOI) RV16-infected BEAS-2B cells. The 50 µL RV16 consistently increased CCL5 production in a time-dependent manner, and 10 nM Dex significantly inhibited RV16-induced CCL5 at 48 and 72 h. * *p* < 0.05 and ** *p* < 0.01 vs. medium groups; ^†^ *p* < 0.05 vs. RV16 groups. Results are mean ± SD (*n* = 8, paired Student’s *t*-test).

**Figure 8 pharmaceuticals-17-01554-f008:**
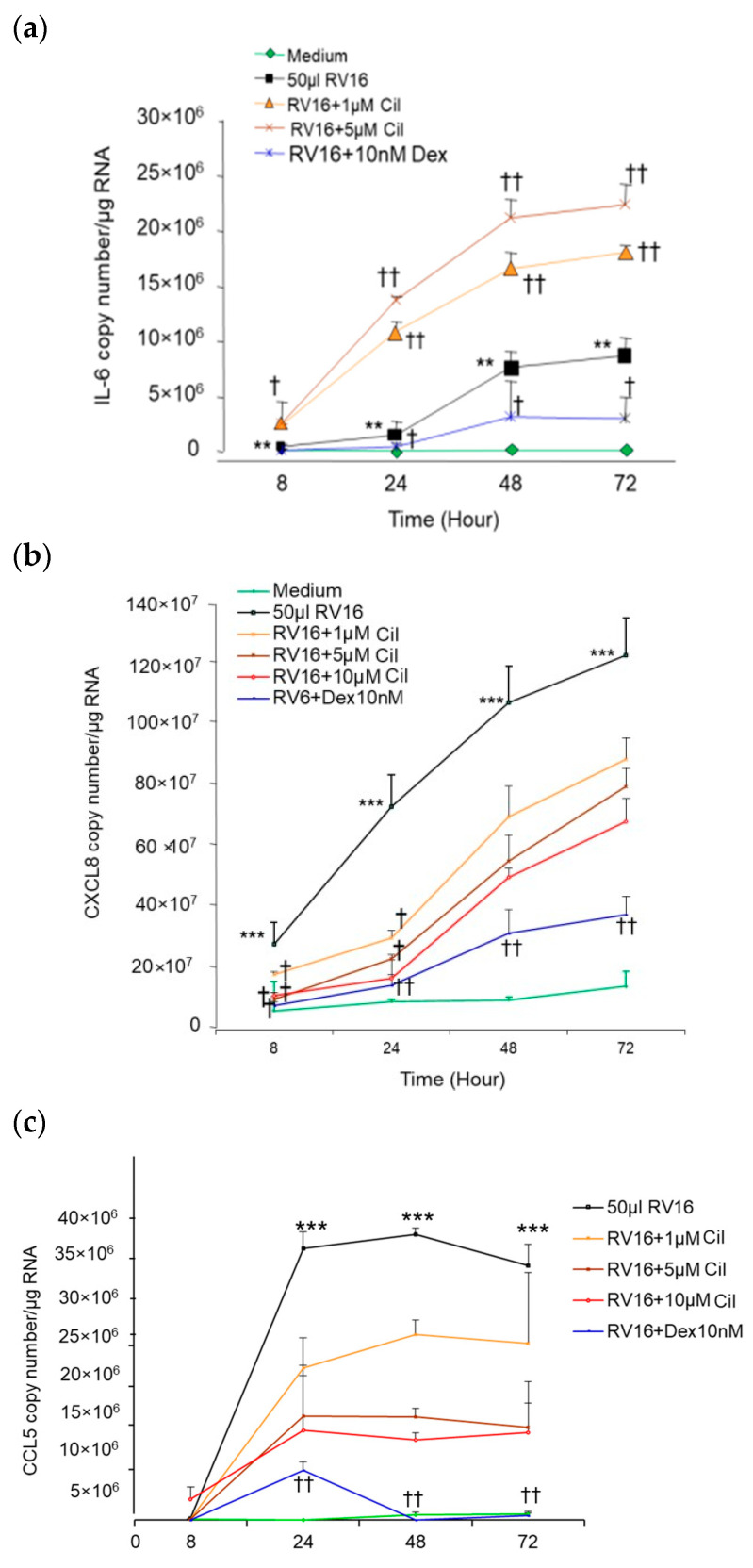
Effect of pre-cilomilast on 50 µL (1 MOI) RV16-induced IL-6, CXCL8, and CCL5 mRNA synthesis in BEAS-2B cells. (**a**) RV16 up-regulated IL-6 mRNA synthesis was further increased by 1 and 5 µM cilomilast (Cil) in both a time- and drug dose-dependent manner. The 10 nM Dex significantly inhibited RV16-induced IL-6 mRNA copies from 24 to 72 h. (**b**) RV16 increased CXCL8 mRNA synthesis in a time-dependent manner, which was suppressed by 1–10 µM Cil at 8 and 24 h in a dose-dependent manner and by 10 nM Dex from 8 to 72 h. (**c**) RV16 significantly increased CCL5 mRNA production and peaked at 24–48 h. Cil did not significantly inhibit RV16-induced CCL5 mRNA copies. However, 10 nM Dex significantly reduced RV16-induced CCL5 mRNA from 24 to 72 h. ** *p* < 0.01 and *** *p* < 0.001 vs. medium groups; ^†^
*p* < 0.05 and ^††^
*p* < 0.01 vs. RV16 groups. Results are mean ± SD (*n* = 8, paired Student’s *t*-test).

**Table 1 pharmaceuticals-17-01554-t001:** Cell surface immunogold labelling assessed using HR-SEM.

RV16 + Cilomilast							Control		Cilomilast Only	
Groups	Medium	RV16	0.1 μM	1 μM	5 μM	10 μM	TNF-α + IFN-γ	RV16 + 10 nM Dex	1 μM	5 μM
Pre-RV16 M(R)	7 (0–86)	29 ** (0–240)	25 (0–172)	18 ^††^ (0–160)	16 ^††^ (0–158)	16 ^††^ (0–149)	97 ** (18–360)	9 ^††^ (0–98)	4.7 * (0–57)	4.3 * (0–54)
Post-RV16 M(R)	8 (0–92)	28 ** (0–220)	26 (0–158)	25 (0–132)	17 ^††^ (0–92)	15 ^††^ (0–109)		11 ^††^ (0–101)		

Dex = dexamethasone. Results are shown expressed as median (M) and range (R) of immunogold particles per 10 μm^2^ cell surface. * *p* < 0.05 and ** *p* < 0.01 vs. medium; ^††^ *p*< 0.01 vs. RV16-infected group. Results are median (range) (*n* = 300 cells, Mann–Whitney U test).

**Table 2 pharmaceuticals-17-01554-t002:** Primers and probes for quantitative PCR.

	Primers: Sense	Primers: Antisense	Probes
IL-6	5′-CCA GGA GCC CAG CTA TGAAC-3′	5′-CCC AGG GAG AAG GCA ACT G-3′	5′-(6-FAM) CCT TCT CCA CAA GCG CCT TCG GT (Tamra-Q)-3′
CXCL8	5′-CTG GCC GTG GCT CTC TTG-3′	5′-CCT TGG CAA AAC TGC ACC TT-3′	5′-(6-FAM) CAG CCT TCC TGA TTT CTG CAG CTC TGT GT(Tamra-Q)-3′
CCL5	5′-GCA TCT GCC TCC CCA TATC-3′	5′-CAG TGG GCG GGC AATG-3′	5′-(6-FAM) TCG GAC ACC ACA CCC TGC TGCT(Tamra-Q)-3′
18S rRNA	5′-CGC CGC TAG AGG TGA AAT TCT-3′	5′-CAT TCT TGGCAA ATG CTT TCG-3′	5′-(6-FAM) ACC GGC GCA AGACGG ACC AGA(Tamra-Q)-3′

## Data Availability

All data are included within this article.

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
