# Peer review of "Cilomilast Modulates Rhinovirus-Induced Airway Epithelial ICAM-1 Expression and IL-6, CXCL8 and CCL5 Production"

_pharmaceuticals, 2024, doi:10.3390/ph17111554_

Round 1
Reviewer 1 Report
Comments and Suggestions for Authors
This paper investigated the effects of pretreatment or posttreatment of Cilomilast, a selective second-generation PDE4 inhibitor, on rhinovirus 16 (RV16)-induced airway epithelial intercellular adhesion molecule-1 (ICAM-1) and inflammatory cytokine/chemoattractant (IL-6, CXCL8 and CCL5) expressions in BEAS-2B bronchial epithelial cells. The results showed that Cilomilast has differential effects on RV16-induced ICAM-1 and interleukins, i.e. inhibiting virus induced ICAM-1 and CXCL8 expressions while increasing IL-6 production. These results may help to explain the beneficial actions of Cilomilast in vivo. There are some concerns as listed in the following:
(1) This paper title seems to be incomplete. -> expression?
(2) Are there any different effects between pretreatment and posttreatment of Cilomilast?
(3) Typos and others
L12: in vitro and in vivo
*L14: cytokine/chemoattractants: IL-6, CXCL8 and CCL5 in vitro -> expression?
*L47: inter alia,? the expression
*L59: LPS: full name first
L74: on/from
L109: At 90% confluence, BEAS-2B cells,
L185: The ELISA and RT-PCR data were expressed as mean ± SD was calculated in each case.
*L257: *P<0.0001 medium, †P<0.0001 or 0.01 RV16 vs RV16-infected group, -> not clear
*L270: Results are mean ± SD -> repetitive writing
L315: inhibited CCL5 production at both pre-? and post-48 and post-72 h time points
L325: 3.3. effects of Cilomilast o
**L341, 346, 347,348: CCR5-> CCL5
*L349: *P< 0.001 or 0.01? vs medium groups; †P<0.01 or 0.05? vs RV16 groups
**L445: cilomilast significantly reduced baseline CXCL8 mRNA synthesis and protein release at 8 and 24 h drug treatment, respectively, but not at 48 and 72h. -> no data were presented at 24h, 48, 72h in Fig. 6C.
L455: including monocytes macrophages,
L484: This work was supported- by GlaxoSmithKline.
L496: Cyclic ade-nosine-3′,5′-monophosphate -> Cyclic adenosine-3′,5′-monophosphate
**L497: References: (check all) Keep one consistent writing format for the
(1) authors: e.g. Ref.1 vs. Ref 3
(2) title: capital prefix on all words (e.g. Ref. 3) or only on the first word (e.g. Ref. 4)
(3) journal: e.g. Ref. 4 vs. R5 vs. R9; R6 vs. R9
(4) page number: e.g. R6 (786-796) vs. R9 (325-36)
**L507=L508
Comments on the Quality of English LanguageThe English could be improved to more clearly express the research.
Author Response
Dear reviewer,
Please see our one to one responses to your comments attached.
Thanks and kind regards
Jie Zhu

Reviewer 2 Report
Comments and Suggestions for Authors
This is a very good work. But some changes and explanations are required.
1. In this study viral load was NOT measured to check increased or decreased RV-induced infection. Simply based on some biomarker studies we can’t conclude about the viral infection level. Viral load can be measured using some virus specific gene expression also.
2. Why the authors have not used Western blot (since they have the monoclonal antibody) or gene expression studies to show the ICAM expression? The data from gold labeling (as shown in Fig is not conclusive because a lot of non-specific labeling is also observed
3. The information given in Fig 2 is NOT clear. Please explain the meaning of X and Y- axis data presentation. If you consider the median value for Fig 2C and Fig 2D, there is no significant difference between 1 and 5 µM treatment – why?
4. For the gold labelling experiment they have to provide 3 figures and 1 table. Instead, with western blot or immune-fluorescence microscopy, they could have got the ICAM expression data in 1 figure. At least, one of these experiments is required to show the change in ICAM expression.
5. Why they have chosen an MOI of 1:2.5 and not 1:10 or higher? Viral load should be measured for the indicated MOI.
6. The discussion is too broad. It should be simpler to link the importance of the PDE4 inhibition with ICAM expression followed by differential cytokine expression.
7. For example it is written as “In conclusion, the present in vitro data are compatible with the hypothesis that cilomilast decreases RV16-induced expression of ICAM-1 and CXCL-8 whilst increasing levels of RV16-induced bronchial IL-6” – How these things are linked so that it will reduce the viral infection. The question is why there should be an increased level of CXCL-8 but a decreased level of IL-6.
8. There are many published works that show similar data or results.
https://www.atsjournals.org/doi/full/10.1164/rccm.200212-1490OC
https://pmc.ncbi.nlm.nih.gov/articles/PMC5418297/
https://pubmed.ncbi.nlm.nih.gov/33182057/
In what way the existing work is significantly different from these previously published works?
9. For the gene expression studies it will be good to show the bar diagrams of different time points showing the fold change in gene expression for each group.
10. If possible, authors should include colored figures (NOT Black and white) for this manuscript
Author Response
Dear Reviewer,
Please see our one to one responses for your comments attached.
Thanks and kind regards
Jie Zhu

Round 2
Reviewer 1 Report
Comments and Suggestions for Authors
Recheck the following points, especially note the marked**:
L35: l burden[1,2]
L37: asthma[7,8]
L43: (CXCL8)[12,15]
L44: expression[14]
*L99: 1 μM and E. 5 μM -> 1 μM and (e) 5 μM
L105: 7(0-86)
L111: 18(0-160) and 16(0-158) -> 18 (0-160) and 16 (0-158)
L150: IFN-r -> IFN-γ
**L151: Table 2: Are there any different effects between pretreatment and posttreatment of Cilomilast? Add the comment of the following Response in the Results section or Discussion section: There was little or no difference between pre-treatment and post-treatment. But as we described in L130-L131 as ‘A significant reduction at 1 μM cilomilast was detected in the pre, but not in the post groups.’ and in L191-192 as ‘RV16 induced CXCL8 production was significantly down-regulated by 25-36% after pre-1-10 µM and post-5-10 µM cilomilast treatment but only at the 24 h time-point.’ These results demonstrate that pre-treatment is more effective at the lower dosage of cilomilast than post-treatment.
**L153: ††P<0.0001 or 0.01 RV16? vs RV16-infected group, -> in RV+ Cilomilast group only † was appeared but no ††. Usual labeling: †P<0.05, ††P<0.01, †††P<0.001 or *P<0.05, **P<0.01, ***P<0.001. For ††P<0.0001 or 0.01, ††P< 0.01 is enough.
Delete ‡P<0.05 vs medium and change the labeling ‡ to * in Cilomilast only group
L161: (P<0.001, Figure. 4 -> (P<0.001, Figure. 4)
**L170: (P<0.001 or 0.01, Figure 5a) -> delete ‘or 0.01’, see Figure legend
**L171: both pre- and post-cilomilast?? treatment at 1-10μM significantly augmented RV16 induced IL-6 protein production at 24h -> The title of Figure 5: Effect of pre-cilomilast on IL-6 protein release from BEAS-2B cells
**L175: at pre-treated 24 and 48 h??
**L192: after pre-1-10 μM and post-5-10 μM cilomilast??
**L193: but only at the 24 h time-point (P<0.01, Figure 6a). -> Figure 6b
**L196: at both pre- and post-cilomilast?? from 8 to 24h
**L212: both pre- and post- cilomilast?? -> The title of Figure 7: Effect of pre-cilomilast on CCL5 protein production from BEAS-2B cells.
**L214: for both pre- and post-treatment?? at 48 and 72 h
**L229, 232: 1-10 μM cilomilast -> 1 and 5 μM cilomilast
**L232, L250: P<0.01 or <0.05 -> P <0.05 is enough to indicate the significant difference between two groups.
*L461: IL-6 CXCL8 and CCL5 -> IL-6, CXCL8 and CCL5
**L476: Table1: needs to readjust
L492: 10 μm2 -> 10 μm2 (superscript 2)
L499: between multiple groups -> among multiple groups
**L521: Abbreviations: adjust the position of AP-1, CREB, IFN-γ
**L522: References: (check all) It is better to keep one consistent writing format for the
(1) title: capital prefix on all words (e.g. Ref. 1,2,3,10… ) vs. only on the first word (e.g. Ref. 4,-9, 11…), select one or follow the guideline of pharmaceuticals
(2 journal: e.g. Ref. 5 (italic letter) vs. Ref. 7, 8...
(3) page number: e.g. R6 (786-796), 13 (1219-1229). 15… vs. R8 (325-36),9 (677-83), 10…
Author Response
Thanks for your valuable comments again. Please see our point by point responses attached.

Reviewer 2 Report
Comments and Suggestions for Authors
The authors should be congratulated on their excellent work.
Viral load should be measured in the future when viral infection studies are done.
Author Response
Thanks, we have added your comment into our discussion, please se L380-390.